# Prevalence and Effect of Low Skeletal Muscle Mass among Hepatocellular Carcinoma Patients Undergoing Systemic Therapy: A Systematic Review and Meta-Analysis

**DOI:** 10.3390/cancers15092426

**Published:** 2023-04-23

**Authors:** Meng-Hsuan Kuo, Chih-Wei Tseng, Ching-Sheng Hsu, Yen-Chun Chen, I-Ting Kao, Chen-Yi Wu, Shih-Chieh Shao

**Affiliations:** 1Department of Pharmacy, Dalin Tzu Chi Hospital, Buddhist Tzu Chi Medical Foundation, Chia-Yi 62247, Taiwan; danms0521@gmail.com (M.-H.K.);; 2School of Medicine, Tzu Chi University, Hualien 97004, Taiwan; 3Division of Gastroenterology, Department of Internal Medicine, Dalin Tzu Chi Hospital, Buddhist Tzu Chi Medical Foundation, Chia-Yi 62247, Taiwan; 4School of Post-Baccalaureate Chinese Medicine, Tzu Chi University, Hualien 97004, Taiwan; 5Department of Pharmacy, Keelung Chang Gung Memorial Hospital, Keelung 20401, Taiwan

**Keywords:** hepatocellular carcinoma, systemic therapy, sarcopenia, survival, low muscle mass, meta-analysis

## Abstract

**Simple Summary:**

The association between low skeletal muscle mass (LSMM) and survival in HCC patients receiving systemic therapy remains inconclusive based on previous studies. Our study aimed to use meta-analysis to aggregate a large sample size and identify the association. The results confirmed that LSMM is prevalent among HCC patients undergoing systemic therapy and is associated with poorer overall survival and progression-free survival. This finding highlights the importance of evaluating muscle mass and early interventions to improve the survival of advanced HCC patients in clinical practice.

**Abstract:**

Low skeletal muscle mass (LSMM) is associated with poor outcomes in hepatocellular carcinoma (HCC) patients. With the emergence of new systemic therapeutics, understanding the effect of LSMM on HCC treatment outcomes is critically important. This systematic review and meta-analysis investigates the prevalence and effect of LSMM among HCC patients undergoing systemic therapy as reported in studies identified in searches of the PubMed and Embase databases published through 5 April 2023. The included studies (*n* = 20; 2377 HCC patients undergoing systemic therapy) reported the prevalence of LSMM assessed by computer tomography (CT) and compared the survival outcomes [overall survival (OS) or progression-free survival (PFS)] between HCC patients with and without LSMM. The pooled prevalence of LSMM was 43.4% (95% CI, 37.0–50.0%). A random-effects meta-analysis showed that HCC patients receiving systemic therapy with comorbid LSMM had a lower OS (HR, 1.70; 95% CI, 1.46–1.97) and PFS (HR, 1.32; 95% CI, 1.16–1.51) than did those without. Subgroup analysis according to systemic therapy type (sorafenib, lenvatinib, or immunotherapy) yielded similar results. In conclusion, LSMM is prevalent among HCC patients undergoing systemic therapy and is associated with poorer survival. Early intervention or prevention strategies to improve muscle mass may be necessary for this patient population.

## 1. Introduction

Liver cancer is the fourth leading cause of cancer-related deaths globally, with hepatocellular carcinoma (HCC) being the most common form [1,2]. Identifying risk factors for the poor outcomes of HCC is crucial for choosing effective treatments and improving survival rates. Advanced HCC is particularly aggressive and has a poor prognosis, making risk stratification even more critical [1,3]. The long-term prognosis for patients with HCC is heavily influenced by liver reserve, cancer staging, and patient performance [4,5]. Unfortunately, the current systems used for determining cancer stage and prognosis lack parameters that consider nutritional, functional, and performance status [5,6]. One such factor is sarcopenia, which independently predicts poor overall survival (OS) and progression-free survival (PFS) in HCC patients [7,8].

Sarcopenia is characterized by the loss of muscle strength (handgrip strength), skeletal muscle mass (muscle index), and physical performance (walking speed) [9]. Numerous methods are available for assessing sarcopenia, with computer tomography (CT) imaging being one of those most commonly used in research. This method is preferred due to its objective nature and widespread availability [10]. CT-detected low skeletal muscle mass (LSMM) has been found to be a predictor of poor prognosis in HCC patients, as demonstrated in previous meta-analyses [7,8]. Although most studies have focused on patients receiving sorafenib, the association between LSMM and survival in HCC patients undergoing systemic therapy has been also established [11,12,13,14,15,16,17]. A recent meta-analysis of 11 studies, including 1148 HCC patients, found that LSMM was associated with increased mortality and decreased time to treatment failure in patients treated with sorafenib or lenvatinib [18].

Several new systemic treatments, including atezolizumab plus bevacizumab, sorafenib, lenvatinib, regorafenib, cabozantinib, and ramucirumab, have shown promise in extending survival in patients with advanced HCC [4]. Earlier studies investigated the effect of LSMM on survival in patients undergoing HCC treatment. Among patients receiving immunotherapy [11,19,20,21,22,23], most studies report no significant association between LSMM and survival [20,24,25], with the exception of two studies [19,26]. Therefore, an updated review and meta-analysis of this topic is needed to gain a better understanding of the relationship between LSMM and the outcomes of HCC patient undergoing a variety of systemic therapy regimes.

## 2. Materials and Methods

This systematic review and meta-analysis was pre-registered on INPLASY (registration number: INPLASY202320011) and followed the Preferred Reporting Items for Systematic Reviews and Meta-Analyses statement (PRISMA) (Appendix A) [27].

### 2.1. Search Strategy

We searched PubMed and Embase for published studies addressing the prevalence of LSMM and its clinical effects on treatment outcomes in HCC patients undergoing systemic therapy, including studies published up to 5 April 2023 [28,29]. We used a free-text search with appropriate MeSH or Emtree terms related to LSMM and liver cancer. Additional articles were identified in the reference lists of pertinent original studies and relevant reviews. No language restrictions were applied to this search. The detailed search strategy is presented in Appendix A.

### 2.2. Inclusion Criteria

After removing duplicate records from different databases, two reviewers (C.-W.T. and M.-H.K.) independently selected the included studies based on the following parameters: (1) Patients: HCC patients treated with systemic therapy (i.e., lenvatinib, sorafenib, and immunotherapy) [27]; (2) Exposures: LSMM; (3) Comparison: non-LSMM; (4) Outcome: prevalence of LSMM, OS, and PFS; (5) Study design: cohort or cross-sectional studies. We excluded duplicate studies from an overlapping population with a smaller sample size and time span [28].

### 2.3. Literature Selection and Data Extraction

Two reviewers (C.-W.T. and M.-H.K.) individually screened titles and abstracts based on the inclusion criteria and read the full-text articles for final eligibility. The agreement between two reviewers was 87% and 92% in these processes, respectively. To resolve discrepancies in the study selection, a third reviewer (C.-S.H.) was consulted to make the final decisions.

Two authors (C.-W.T. and M.-H.K.) independently collected data, including the name of the first author, publication year, country, setting, study design, treatment regimens, population, number of patients, sex ratio, age, method used for estimating muscle mass, LSMM cut-off value, the prevalence of LSMM, study period, and statistical data on the influence of LSMM on OS and PFS (including adjustment factors). All disagreements were resolved through discussions.

### 2.4. Assessment of Methodological Quality

The methodological quality of the included studies was assessed by two researchers (C.-W.T. and M.-H.K.) independently. The quality of the involved studies was evaluated using the Newcastle–Ottawa Scale (NOS) [30]. The studies for which the researchers differed in their assessment of quality were resolved through discussion.

### 2.5. Statistical Analysis

The random-effects meta-analysis of single proportions was used to estimate the pooled prevalence of LSMM in HCC patients undergoing systemic therapy. The pooled OS and PFS were compared between these patients with and without LSMM using the adjusted HR (or unadjusted HR for studies that did not report the adjusted HR) and 95% CI using a random-effects meta-analysis model. Subgroup analyses were conducted on groups defined by treatment regimen (lenvatinib, sorafenib, or immunotherapy), study region (Asian vs. non-Asian area), method used to estimate muscle mass [skeletal muscle index (SMI) vs. psoas muscle index (PMI)], and study quality (good vs. poor). To address the survival of patients with advanced HCC, we conducted one additional subgroup analysis based on observational time (over vs. under 2 years) in the OS and PFS outcomes [4]. We used I^2^ to measure the statistical heterogeneity among the included studies and funnel plots/Egger’s test to determine the potential publication bias. Two-sided *p* < 0.05 was considered statistically significant, and all calculations were performed using Comprehensive Meta-Analysis version 4.0.

## 3. Results

### 3.1. Literature Search and Study Selection

We identified 1232 relevant published studies. After excluding 361 duplicated studies, a total of 871 studies were screened based on titles and abstracts, leaving 166 full-text articles for further assessment. The inclusion criteria were not met by 146 of these articles, as follows: 30 studies on hepatic tumors other than HCC; 3 studies with analysis not related to sarcopenia or muscle mass; 99 studies with patients receiving a therapy other than the recommended systemic therapy; 12 studies lacking statistical data on the influence of pretreatment LSMM on OS or PFS; and 2 studies with overlapping patient data. The reasons for exclusion in the final stage are shown in Appendix A. The flow diagram for this study selection process is shown in Figure 1.

### 3.2. Characteristics of Included Studies

The meta-analysis included 20 retrospective studies involving a total of 2377 patients with HCC who received systemic therapy [11,12,14,15,16,17,19,20,21,22,23,24,25,26,31,32,33,34,35,36]. One study reported the prevalence of LSMM and survival analysis results separately for lenvatinib and immunotherapy [20], while another conducted survival analysis separately for males and females [16]. All studies except one reported the OS [21]. Thus, a total of 21 prevalence records [11,12,14,15,16,17,19,20,21,22,23,24,25,26,31,32,33,34,35,36], 20 OS analyses [11,12,14,15,16,17,19,20,22,23,24,25,31,32,33,34,35,36], and 13 progression-free survival analyses were used in the meta-analysis (Table 1) [11,15,19,20,21,23,24,25,26,33,35,36]. The sample size of each study cohort ranged from 32 to 356. Most of the patients were male (*n* = 1930; 81%), and one study [11] enrolled only males. Of the 20 studies included, 17 were from Asian countries [11,12,15,16,17,21,22,23,24,26,31,33,34,35] and 3 were from non-Asian countries [14,25,32]. The HCC treatments used included sorafenib (*n* = 11) [11,13,14,15,16,17,31,32,33,35,36], lenvatinib (*n* = 5) [12,20,22,23,34], and immunotherapy (*n* = 6) [19,20,21,24,25,26]. The measured methods and cutoff values for muscle mass varied between studies. Seventeen records used the SMI [11,14,15,16,19,20,21,23,24,25,26,31,32,33,34,35], while four used the PMI [12,17,22]. The median observation period was 3.8 years (range, 0.7–10 years), and 16 (76%) of the records had an observation period of more than 2 years, which was the expected survival time for advanced HCC patients [4]. Of these studies, two were found to have poor methodological quality, while 17 were deemed to have good quality. The risk of bias in the included studies is outlined in Appendix A.

### 3.3. Prevalence of LSMM among HCC Patients Undergoing Systemic Therapy

The prevalence of LSMM was reported in 21 studies, including 2377 individuals [11,12,14,15,16,17,19,20,21,22,23,24,25,26,31,32,33,34,35,36]. The range of prevalence was 15–65.1%. The overall pooled prevalence was 43.4% (95% CI, 37.0–50.0%; I^2^, 89.27%; *p* < 0.001; Egger’s test, *p* = 0.10) (Figure 2). The sub-analysis of records from Asian individuals revealed a prevalence of 42.2% (95% CI, 34.7–50.1%). Among non-Asian individuals, the prevalence was 51.0% (95% CI, 46.3–55.7%). The prevalence was 44.5% (95% CI, 37.7–51.6%) among studies that defined LSMM using the L3-SMI and 38.9% (95% CI, 22.7–58.0%) among studies using L3-PMI (Appendix A).

### 3.4. Overall Survival among HCC Patients Undergoing Systemic Therapy with Versus without LSMM

Twenty records [11,12,14,15,16,17,19,20,22,23,24,25,31,32,33,34,35,36] including 2248 patients reported OS results. The pooled HR was 1.70 (95% CI, 1.46–1.97; *p* < 0.001), without significant heterogeneity among the included studies (I^2^, 29.27%; Egger’s test, *p* = 0.20) (Figure 3A). The subgroup analysis results are shown in Appendix A. The crude and adjusted pool analyses showed the same positive association between LSMM and poor prognosis. The crude HR pooling from 16 records with univariate analysis results was 1.68 (95% CI, 1.44–1.95; *p* < 0.001; I^2^, 24.37%), and the adjusted HR pooling from 16 records with multivariate analysis data was 1.84 (95% CI, 1.59–2.13; *p* < 0.001; I^2^, 11.07%). LSMM was consistently associated with a poor OS in the subgroup analysis of sorafenib (HR, 1.74; 95% CI, 1.41–2.14; *p* < 0.001), lenvatinib (HR, 1.71; 95% CI, 1.22–2.41; *p* = 0.002), and immunotherapy (HR, 1.61; 95% CI, 1.15–2.24; *p* = 0.005) (Figure 3B–D). The pooled data from records of Asian (HR, 1.80; 95% CI, 1.54–2.11; *p* < 0.001), non-Asian (HR, 1.31; 95% CI, 1.07–1.62; *p* = 0.010), SMI (HR, 1.74; 95% CI, 1.44–2.10; *p* < 0.001), and PMI (HR, 1.61; 95% CI, 1.22–2.11; *p* = 0.001) demonstrated the same association. In the pooled HR from studies that observed patients for more than 2 years, the association remained significant (HR, 1.69; 95% CI, 1.46–1.96; *p* < 0.001). However, this was not the case for studies with an observation time under 2 years (HR, 1.94; 95% CI, 0.54–6.90; *p* = 0.308) (Appendix A).

### 3.5. Progression-Free Survival among HCC Patients Undergoing Systemic Therapy with and without LSMM

Upon further examination of PFS, we included 13 records involving 1400 patients [11,15,19,20,21,23,24,25,26,33,35,36]. A meta-analysis revealed a pooled HR of 1.32 (95% CI, 1.16–1.51; *p* < 0.001) (Figure 4A). The findings of the subgroup analysis are shown in Appendix A. The crude HR pooled from nine records with univariate analysis results was 1.57 (95% CI, 1.24–1.98; *p* < 0.001), and the adjusted HR pooled from seven records with multivariate analysis data was 1.32 (95% CI, 1.10–1.59; *p* = 0.003). A correlation between PFS and LSMM was identified in the subgroup analysis of treatment type for sorafenib (HR, 1.23; 95% CI, 1.03–1.46; *p* = 0.020), lenvatinib (HR, 2.08; 95% CI, 1.18–3.67; *p* = 0.012), and immunotherapy (HR, 1.41; 95% CI, 1.12–1.78; *p* = 0.004) (Figure 4B–D). The pooled data from records of Asian patients (HR, 1.34; 95% CI, 1.17–1.54; *p* < 0.001) and subgroups by study quality and observation time, demonstrated a similar correlation.

### 3.6. Publication Bias and Sensitivity Analysis

The symmetrical distribution in the funnel plot assessing prevalence suggests the absence of publication bias (Egger’s test, *p* = 0.10) (Appendix A). The funnel plot for assessing the association between LSMM and OS or PFS was visually symmetrical (Appendix A); this result was confirmed by the Egger’s test (*p* = 0.20 and *p* = 0.10, respectively). A sensitivity analysis using the one-study removal method showed a consistently statistically significant effect of LSMM on OS and PFS (Appendix A). The summarized effect sizes did not change the significance of these findings upon removal of any of the included studies, indicating that the pooled results were robust.

## 4. Discussion

This systematic review and meta-analysis involved 20 retrospective studies and 2377 patients. Our results demonstrate that LSMM is a common occurrence among HCC patients receiving systemic therapy, with a prevalence of 43.4%. Our findings indicate that low LSMM is associated with a reduced OS. This correlation was observed across different treatment types (such as sorafenib, lenvatinib, and immunotherapy), geographical regions (Asian and non-Asian), and measurement methods (SMI and PMI). Despite previous studies reporting no correlation between LSMM and PFS [11,15,19,20,21,23,24,25,33], our meta-analysis showed that LSMM is indeed associated with poor PFS among patients undergoing several types of systemic therapy.

Patients undergoing systemic therapy for HCC often have LSMM, with a prevalence ranging from 15 to 65.1% in the studies included in this meta-analysis. The cumulative prevalence in this population was 43.4% (95% CI, 37.0–50.0%), which is similar to that found in a previous meta-analysis of HCC patients treated with sorafenib or lenvatinib (41%, based on data from 10 studies and 1028 patients) [18]. Compared to the overall HCC population, the prevalence of LSMM was higher among those undergoing systemic therapy. For example, a previous meta-analysis by Chuan et al. that included 42 studies with 8203 patients found that the pooled prevalence of sarcopenia among HCC patients was 39% (95% CI, 33–45%; *n* = 8203) [7]. Another meta-analysis by March et al. including 25 studies reported a cumulative prevalence of 38.5% for LSMM in patients with HCC [8]. Patients with advanced HCC (BCLC stage C) or cancer progression to an advanced stage typically undergo systemic treatment as the standard of care [1,3,4]. These patients often have sarcopenia as a result of the tumor burden and systemic cytokine-mediated inflammation caused by the cancer [10,37]. Furthermore, disease progression and previous cancer treatments can lead to cachexia, a condition that causes loss of muscle and weight [38]. With such a high prevalence, LMSS seems to be an important issue in this population.

Multiple meta-analyses have demonstrated an association between sarcopenia and poor prognosis for cancers such as lung, breast, and lymphoma [39,40,41], although the mechanism underlying this effect is unclear. Previous meta-analyses have also demonstrated that LSMM affects both OS and PFS in patients with HCC [7,8]. Our study, which focused on patients receiving systemic therapy with CT scans used to measure muscle mass, found that LSMM was also associated with significantly reduced OS and PFS. In summary, the presence of LSMM was found to have a negative effect on OS and PFS among HCC patients receiving systemic therapy. Our findings suggest that LSMM may be a valuable tool for guiding treatment decisions and improving patient outcomes in the management of HCC. Further research is needed to confirm these results and to determine the optimal therapeutic approach for patients with LSMM.

The previous meta-analyses only showed that LSMM is a predictor of OS in patients receiving kinase inhibitors, such as sorafenib or lenvatinib [8,18]. The subgroup analysis in our study provided stronger evidence for the effect of LSMM on OS and also on PFS in patients receiving the same treatments. The association between LSMM and poor survival in patients treated with kinase inhibitors may result from sarcopenia patients often receiving lower initial doses of medication and experiencing more dose-limiting toxicity [10,42]. Both treatments have been found to result in grade 3–4 drug-related adverse events in about 50% of treated patients, resulting in a withdrawal rate near 15% [1]. Studies have suggested that sarcopenia can decrease a patient’s ability to tolerate medication [12,14,15]. Additionally, a direct pathway may be involved, such as changes in the phosphatidylinositol-3-kinase/AKT–mammalian target of rapamycin pathway, which is essential for muscle protein synthesis [5,43]. Dysregulation of this pathway caused by medication may lead to further muscle loss [44]. These results suggest that LSMM should be considered as a factor in decision making for protein kinase inhibitor treatment in HCC patients.

No meta-analysis has addressed the association between LSMM and the effectiveness of immunotherapy in patients with HCC. The major studies have found no significant correlation between LSMM and survival (OS and PFS) in patients receiving immunotherapy [20,24,25]. Two studies found a correlation between LSMM and survival [19,26]. After pooling these studies, this meta-analysis revealed that patients receiving immunotherapy who had LSMM had a shorter OS and PFS than did those without LSMM. Although the relationship between LSMM and reduced effectiveness of immunotherapy is unclear, it may be explained by the tumor microenvironment (inflammation and immunity) and cytokine activity. Skeletal muscle is now considered an organ with immune regulatory properties [45]. It produces and releases important proteins known as myokines, which counteract the harmful effects of proinflammatory adipokines and contribute to the tumor microenvironment [10,45,46,47]. However, when muscle wasting occurs, these counteracting effects disappear, resulting in systemic inflammation and immune disturbances [47]. Studies have shown that inflammatory factors and LSMM are associated with survival in HCC patients receiving immunotherapy [20,21,24]. On the other hand, skeletal muscle cells modulate immune function by signaling through different soluble factors, cell surface molecules, or cell-to-cell interactions [45]. The interaction between muscle wasting and immune senescence appears to be bidirectional. Cytokines such as transforming growth factor-β [48] and interleukin-6 [49] may play a role in the development of sarcopenia and contribute to T-cell exhaustion, which can lead to a poor response to immunotherapy [10,50]. Skeletal muscle depletion also causes a decrease in myokine levels, which may result in a poor response to immunotherapy [45,46]. Therefore, LSMM may affect immune regulation and increase resistance to immunotherapy, leading to poorer outcomes among HCC patients.

The key strength of this study is the use of updated data from a variety of studies that included HCC patients receiving systemic therapy. These additional data increased the statistical power of this study, enabling subgroup analyses of the effect of LSMM on OS and PFS. As a result, this study can aid physicians in understanding the effect of LSMM in patients receiving immunotherapy. Furthermore, while most previous studies reported no association between LSMM and progression-free survival among HCC patients receiving systemic therapy, our meta-analysis revealed a correlation due to the larger sample size polled. Third, this is the first meta-analysis to address the association between LSMM and the effectiveness of immunotherapy in patients with HCC. Further research is needed to better understand the underlying mechanisms and potential interventions.

However, we acknowledged several limitations in this systematic review and meta-analysis. First, the variation between studies in the methods and cutoff values used to determine LSMM through CT images may have resulted in the varying prevalence rates. However, the subgroup analyses suggests that the association between LSMM and survival is not affected by these differences, indicating that LSMM has a negative effect regardless of method and cutoff value. For future research, it is important to use standardized assessment methods and cutoff values, as well as conducting prospective studies. Second, most studies only included Asian patients, primarily from Japan, leading to uncertainty about the applicability of these findings to patients from other regions. Additional studies that include patients from other regions are needed to address this limitation. Third, the patient cohorts used to study sarcopenia among HCC patients vary with respect to etiology and Child–Pugh scores. To obtain a more precise evaluation of the effect of sarcopenia on mortality, a new prediction system that considers LSMM is needed. Fourth, some included studies in this meta-analysis had small sample sizes, but the results from these included studies were consistent with the pooled results of the meta-analysis. In addition, the sensitivity analysis using the one-study removal method indicated that the pooled results were robust. Therefore, the impacts from the inclusion of small sample sizes on our findings were relatively minor.

## 5. Conclusions

LSMM is prevalent among HCC patients undergoing systemic therapy with drugs such as sorafenib, lenvatinib, and new immunotherapeutic agents and is associated with poor OS and PFS. Early intervention or prevention strategies to improve muscle mass may be necessary for this patient population.

## Figures and Tables

**Figure 1 cancers-15-02426-f001:**
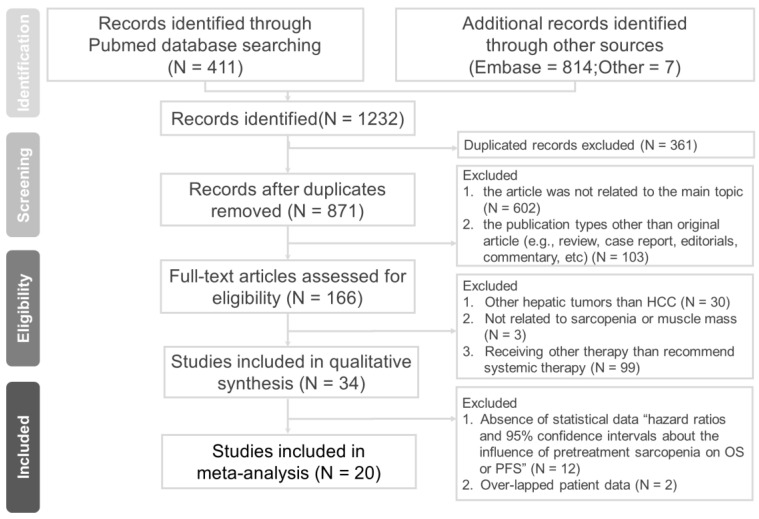
PRISMA diagram of study selection.

**Figure 2 cancers-15-02426-f002:**
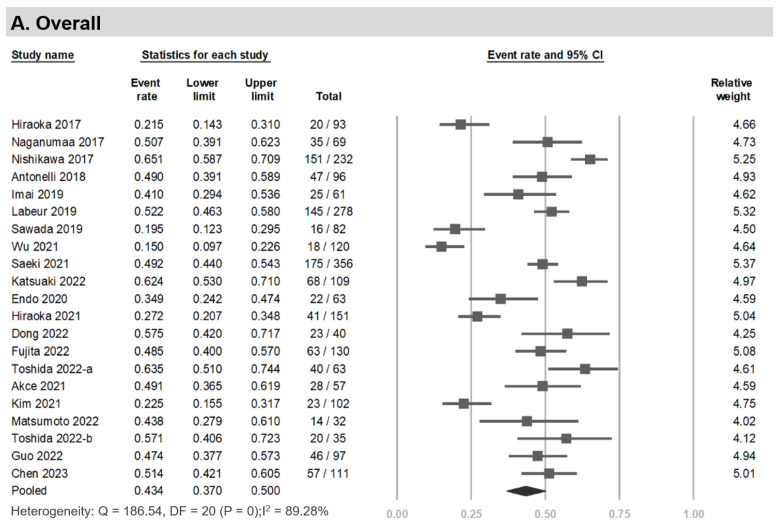
Prevalence of LSMM in HCC patients undergoing systemic therapy. (**A**) Overall, (**B**) sorafenib, (**C**) lenvatinib, (**D**) immunotherapy [11,12,14,15,16,17,19,20,21,22,23,24,25,26,31,32,33,34,35,36].

**Figure 3 cancers-15-02426-f003:**
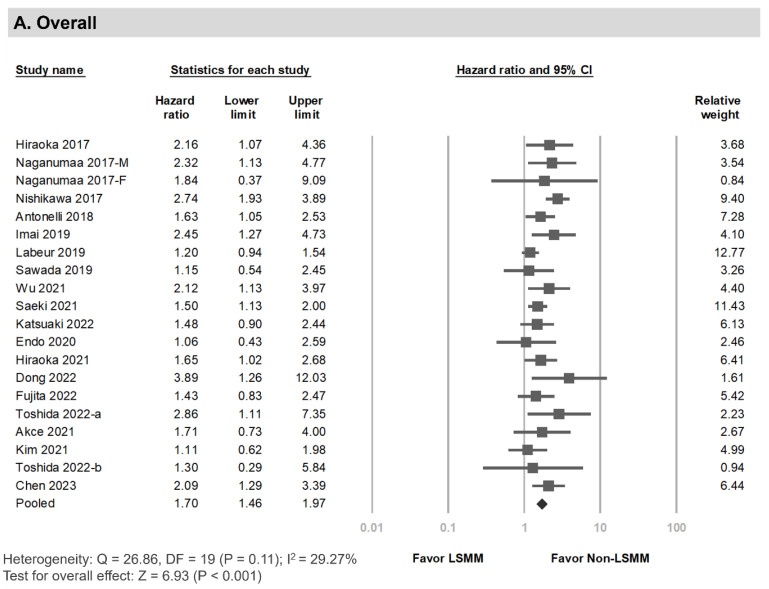
Forest plots of the association between LSMM and overall survival. (**A**) Overall, (**B**) sorafenib, (**C**) lenvatinib, (**D**) immunotherapy [11,12,14,15,16,17,19,20,22,23,24,25,26,31,32,33,34,35,36].

**Figure 4 cancers-15-02426-f004:**
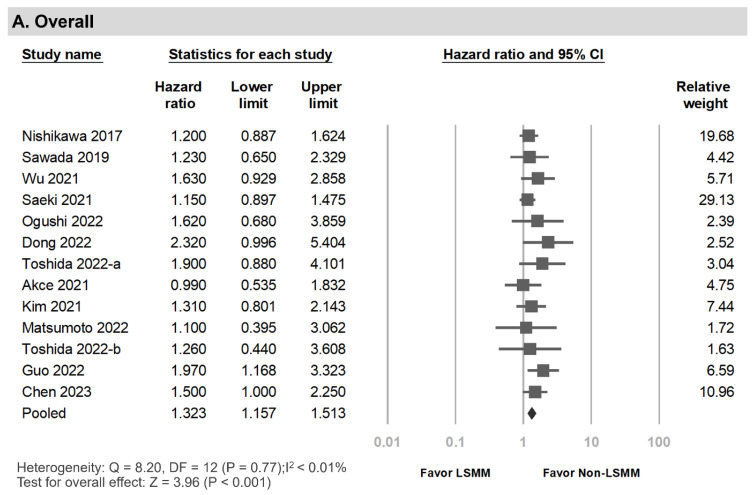
Forest plots of the association between LSMM and progression-free survival. (**A**) Overall, (**B**) sorafenib, (**C**) lenvatinib, (**D**) immunotherapy [11,15,19,20,21,23,24,25,26,33,35,36].

**Table 1 cancers-15-02426-t001:** Demographic and characteristics of included study cohorts.

First Author (Year)	Country	Setting	Treatment Regimen	Patients, *n* (Male/ Female)	Age, Years	Method Used to Estimate Muscle Mass	Cut-off Value for Pretreatment LSMM	LSMM (%) Yes/No	Study Period (Year)	Into OS MA	OS Adjustment Factors	OS HR (95% CI)	Into PFS MA	PFS Adjustment Factors	PFS HR (95% CI)
**Sorafenib**															
Hiraoka (2017) [17]	Japan	multi-center	sorafenib 800/400/200 mg/day	93 (81/12)	68.3 #	L3-PMI	M: 4.24 cm^2^/m^2^; F: 2.50 cm^2^/m^2^	21.5% (20/73)	4	Yes	age, sex, DCP > 100 mAU/mL, positive for bone metastases	2.16 (1.07–4.36)	NR	NR	NR
Nishikawa (2017) [15]	Japan	multi-center	sorafenib 800 mg/day	232 (181/51)	72 *	L3-SMI	M: 36.2 cm^2^/m^2^; F: 29.6 cm^2^/m^2^	65.1% (151/81)	5.5	Yes	age, sex, initial dose, ECOG-PS extrahepatic metastases, portal vein invasion, tumor burden ≥ 50%, AST, ALP, ascites, serum albumin level, serum AFP, DCP	2.74 (1.92–3.92)	Yes	univariate	1.20 (0.89–1.63)
Naganumaa (2017-M) (2017-F) [16]	Japan	single-center, Takasaki	sorafenib (100–800 mg/day)	69 (51/18)	72 *	L3-SMI	<42 cm^2^/m^2^	51% (35/34)	5.5	Yes	age, Child–Pugh score, clinical stage, AFP	2.32 (1.13–4.77)	NR	NR	NR
Antonelli (2018) [14]	Rome	multi-center	sorafenib	96 (75/21)	69 #	L3-SMI	M: BMI > 25 : 53cm^2^/m^2^ BMI < 25: 43 cm^2^/m^2^; F: 41 cm^2^/m^2^	49% (47/49)	2.1	Yes	age, sex, vascular invasion, MELD score,	1.63 (1.05–2.53)	NR	NR	NR
Imai (2019) [31]	Japan	single-center, Gifu	sorafenib 800 mg/day	61 (54/7)	67.3 #	L3-SMI	M: 42 cm^2^/m^2^; F: 38 cm^2^/m^2^	41% (25/36)	4.1	Yes	age, sex, L3SMI, DSFMI, therapeutic effect	2.45 (1.27–4.73)	NR	NR	NR
Labeur (2019) [32]	Netherlands	multi-center	sorafenib	278 (220/58)	64 *	L3-SMI	M: 52.4 cm^2^/m^2^; F: 38.5 cm^2^/m^2^	52% (145/133)	4	Yes	univariate	1.20 (0.94–1.54)	NR	NR	NR
Sawada (2019) [33]	Japan	single-center, Asahikawa	sorafenib	82 (67/15)	69 #	L3-SMI	M: 36.2 cm^2^/m^2^; F: 29.6 cm^2^/m^2^	20% (16/66)	4.1	Yes	age, sex, AFP ≥ 100 ng/mL, BCLC stage C, additional/subsequent therapies, low skeletal muscle mass, positive invasion of hepatic vessels, duration of Sorafenib treatment	1.15 (0.54–2.47)	Yes	univariate	1.23 (0.65–2.33)
Wu (2021) (males only) [11]	Taiwan	single-center, Taipei	first-line sorafenib-containing therapy or combined with tegafur/uracil	120	NA	L3-SMI	39.1 cm^2^/m^2^	15% (18/102)	10	Yes	age, LSMM of TSM; elderly; underweight; HBsAg; anti-HCV; ALBI grade 2; AFP ≥ 400 ng/mL; macrovascular invasion; extrahepatic metastasis; BCLC C; CLIP score ≥ 3; ECOG PS ≥ 1; combination therapy (vs. sorafenib alone)	2.12 (1.13–3.97)	Yes	age, sex, body weight, HBsAg, HCV, ALBI group, AFP, macro vascular invasion, extrahepatic metastasis, BCLC C, CLIP score, ECOG PS, combination therapy	1.63 (0.93–2.86)
Saeki (2021) [35]	Japan	multi-center	sorafenib 800 mg/day	356 (287/69)	69.5 *	L3-SMI	M < 45 cm^2^/m^2^; F < 38 cm^2^/m^2^	49% (175/181)	7.5	Yes	age, sex, BMI, ECOG-PS, Child–Pugh class, tumor number, tumor size, macrovascular invasion, extrahepatic spread	1.50 (1.13–2.00)	Yes	age, sex, BMI, ECOG-PS, Child–Pugh class, tumor number, tumor size, macrovascular invasion, extrahepatic spread	1.15 (0.90–1.48)
Ogushi (2022) [36]	Japan	single-center, Yokohama	sorafenib 800/400 mg/day	109 (84/25)	73 *	L3-PMI	M: 7.038 cm^2^/m^2^; F: 4.400 cm2/m2	62% (68/41)	6.7	Yes	age, sex, HCV or HBV, BMI, Child–Pugh score, PS, BCLC stage, past history of TACE, AFP, DCP	1.48 (0.90–2.45)	Yes	age, sex, HCV or HBV, BMI, Child–Pugh score, PS, BCLC stage, past history of TACE, AFP, DCP	1.62 (0.68–1.86)
**Lenvatinib**															
Endo (2020) [34]	Japan	single-center, Iwate	lenvatinib (8 mg/day < 60 kg or 12 mg/day > 60 kg)	63 (53/10)	71 *	L3-SMI	M < 42 cm^2^/m^2^; F < 38 cm^2^/m^2^	35% (22/47)	0.7	Yes	univariate	1.06 (0.43–2.56)	NR	NR	NR
Hiraoka (2021) [12]	Japan	multi-center	lenvatinib (8 mg/day < 60 kg or 12 mg/day > 60 kg)	151 (116/35)	NA	L3-PMI	M: 4.24 cm^2^/m^2^; F: 2.50 cm^2^/m^2^	27% (41/110)	2	Yes	age, sex, AFP, BCLC stage (C and D), BMI	1.65 (1.02–2.69)	NR	NR	NR
Dong (2022) [23]	China	single-center, Changchun	lenvatinib (8 mg/day < 60 kg or 12 mg/day > 60 kg)	40 (37/3)	59 *	L3-SMI	M: 42 cm^2^/m^2^; F: 38 cm^2^/m^2^	57.5% (23/17)	0.8	Yes	age, Alb, maximum tumor diameter, portal vein thrombosis	3.89 (1.26–12.05)	Yes	univariate	2.32 (1.00–5.41)
Fujita (2022) [22]	Japan	multi-center	lenvatinib 4 mg/8 mg/12 mg based on their body weight and liver function reserve	130 (107/23)	70 *	L3-PMI	M: 6 cm^2^/m^2^; F: 3.4 cm^2^/m^2^	48% (63/67)	2.5	Yes	univariate	1.43 (0.83–2.47)	NR	NR	NR
Toshida (2022-a) [20]	Japan	single-center, Fukuoka	lenvatinib (8 mg/day < 60 kg or 12 mg/day > 60 kg	63 (43/20)	69–75 *	L3-SMI	M: 42 cm^2^/m^2^; F: 38 cm^2^/m^2^	68.2% (40/23); ATZ/BEV, 57.1% (20/15); LEN, 63.5% (40/23)	3.8	Yes	age, sex, LMR < 4.0, ALBI grade, best response	2.86 (1.11–7.33)	Yes	ALBI grade, 2/3 (vs. 1)	1.90 (0.88–4.10)
**Immunotherapy**															
Akce (2021) [25]	Georgia	single-center	anti-PD1 antibody-containing regimens	57 (44/13)	66 *	L3-SMI	M: 43 cm^2^/m^2^; F: 39 cm^2^/m^2^	49.1% (28/29)	3	Yes	age, sex, BCLC stage (B and C vs. A); Inflammation biomarkers	1.71 (0.73–4.00)	Yes	sex, Child–Pugh score, inflammation biomarkers	0.99 (0.54–1.85)
Kim (2021) [24]	Korea	single-center, Seoul	intravenous nivolumab 3 mg/kg	102 (87/15)	61.3 *	L3-SMI	M: 42 cm^2^/m^2^; F: 38 cm^2^/m^2^	22.5% (23/79)	2	Yes	age, sex, ECOG PS, ALBI group, AFP, intrahepatic tumor burden, surgery, RT, ALC and NLR risk group	1.11 (0.62–1.97)	Yes	univariate	1.31 (0.80–2.14)
Matsumoto (2022) [21]	Japan	single-center, Tokyo	ATZ 1200 mg + BEV 15 mg/kg Q3W	32 (19/13)	77 *	L3-SMI	M: 42 cm^2^/m^2^; F: 38 cm^2^/m^2^	53% (17/15)	1.5	NR	NR	NR	Yes	univariate	1.10 (0.40–3.10)
Toshida (2022-b) [20]	Japan	single-center, Fukuoka	ATZ 1200 mg + BEV 15 mg/kg Q3W	35(28/7)	72 *	L3-SMI	M: 42 cm^2^/m^2^; F: 38 cm^2^/m^2^	57.1% (20/15)	3.8	Yes	univariate	1.30 (0.29–5.86)	Yes	A+B+L: ALBI grade, 2/3 (vs. 1), sarcopenia, LEN:ALBI grade, 2/3(vs1), sarcopenia	1.26 (0.44–4.20)
Guo (2022) [26]	China	single-center, Hubei	camrelizumab	97 (79/18)	52 #	L3-SMI	M: 37.7 cm^2^/m^2^; F: 34.3 cm^2^/m^2^	47.4% (46/51)	1.3	NR	NR	NR		number of tumors, Child-Pugh class, macrovascular invasion, extrahepatic spread, ECOG performance, tumor size, PLR, NLR	1.97 (1.17–3.33)
Chen (2023) [19]	Taiwan	single-center, Taipei	immunotherapy	111 (97/14)	59 #	L3-SMI	M: 40.8 cm^2^/m^2^; F: 34.9 cm^2^/m^2^	51.3% (57/54)	5	Yes	Age, sex, multinodular or massive, Child–Pugh, myosteatosis	2.09 (1.29–3.39)	Yes	univariate	1.50 (1.00–2.25)

* Median; # mean.; AFP, alpha-fetoprotein; ALC, absolute lymphocyte count; ALP, alkaline phosphatase; ALT, alanine transaminase; anti-HCV, anti-hepatitis C virus antibody positive; anti-PD1, anti-programmed death-1; AST, aspartate transaminase; ATZ, atezolizumab; BCLC, Barcelona Clinic Liver Cancer; BEV, bevacizumab; BMI, body mass index; CLIP, Cancer of the Liver Italian Program; DCP, deso-protein prothrombin; ECOG-PS, Eastern Cooperative Oncology Group performance; F, female; HBsAg, hepatitis B virus antigen positive; LMR, lymphocyte–monocyte ratio; LSMM, low skeletal muscle mass; M, male; MA, meta-analysis; mALBI grade, modified albumin–bilirubin grade; MELD, Model for End-stage Liver Disease; NA, not available; NLR, neutrophil-to-lymphocyte ratio; NR, HR not reported; OS, overall survival; PFS, progression-free survival; PMI, psoas mass index; PLR, platelet-to-lymphocyte ratio; PS, paraspinal muscle Q3W, every 3 weeks; RT, radiation therapy; SMI, skeletal muscle mass index; TACE, transarterial chemoembolization; TSM, total skeletal muscle; TKIs, tyrosine kinase inhibitors.

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
