# Peer review of "Prevalence and Effect of Low Skeletal Muscle Mass among Hepatocellular Carcinoma Patients Undergoing Systemic Therapy: A Systematic Review and Meta-Analysis"

_cancers, 2023, doi:10.3390/cancers15092426_

Round 1

Reviewer 1 Report

I would like to congratulate the authors on a nice and high quality systematic review with meta analysis.

The main question addressed by the research is:  the effect/influence of low muscle mass in patients with HCC undergoing systemic treatment. This is a relevant topic and gives an update on the current literature in this patient group. Good search and reporting of the studies with their results. They might do an updated search to cover the last time period between November 10, 2022 and now. The conclusions are consistent with the evidence and arguments presented and they address the main question posed. Furthermore, I found the included papers relevant. I would add table 2 as supplement.

Reviewer 2 Report

The authors perform a systematic review analysing the main works available in literature on the effect of low scheletal muscle-mass on survival outcomes of HCC undergoing systemic treatments. The topic is relevant, because the explored issue is still poorly investigated in literature and the metanalysis contributes to create a wider cohort, making possible to obtain a more objective analysis (even if with the limitation of almost exclusively Asian populations). It grants an overall overview of literature available so far about the role of sarcopenia on the outcome of HCC on systemic therapy and the possible therapeutic implications, especially considering the incidence of LSSM in oncologic patients. The metanalisis is correctly developed and the study selection is accurate. The authors highlight the crucial role of LSMM on the outcome of HCC undergoing systemic treatments: they underline the importance of early prevention of sarcopenia, in order to increase treatment efficacy and enhance outcome. References are appropriate and adequately selected. Tables are detailed and well done, providing additional information that enrich the results of this research.

Author Response

Thank you for your diligent efforts and valuable feedback on our article. We hope our manuscript is now appropriate to be accepted for the publication of Cancers after the revisions.  

Reviewer 3 Report

The authors summarized the prevalence and effect of low skeletal-muscle mass (LSMM) among HCC patients undergoing systemic therapy including tyrosine kinase inhibitors and immune checkpoint inhibitors (ICI), with a systematic review and meta-analysis. They concluded that LSMM is associated with poor prognosis in such patients, both in OS and PFS. The paper is generally well written and easy to read, but the conclusion has already been well documented, and I cannot find any novel impact from the study. The only thing might be the relationship between the ICI administration and LSMM in survival, so that the authors should emphasize the poor immune function in LSMM patients with adequate references.

Reviewer 4 Report

The authors provided an updated review and meta-analysis of the relationship between LSMM and the outcome of HCC patient undergoing a variety of systemic therapy regimes, which will generate a wide readership for researchers working in this field. However, there are still some points which need to be fixed before publish.

1) If possible, the authors could include more literatures from all the public sources, such as Cochrane and Web of Science, beyond Pubmed and Embase databases.

2) How about the latest research articles published during 11/2022-3/2023? If not included, at least discussed.

3) The details of the mouse  or patients samples included in the analyzed studies should be considered as a factor for exclusion.

4) The figure quality needs to be further improved, such as Figure 2. Diverse displaying formats are highly recommended to attract readers' interest.

Round 2

Reviewer 4 Report

The authors have answered all my questions.